# Low Trapping Effects and High Electron Confinement in Short AlN/GaN-On-SiC HEMTs by Means of a Thin AlGaN Back Barrier

**DOI:** 10.3390/mi14020291

**Published:** 2023-01-22

**Authors:** Kathia Harrouche, Srisaran Venkatachalam, Lyes Ben-Hammou, François Grandpierron, Etienne Okada, Farid Medjdoub

**Affiliations:** CNRS-IEMN, Institute of Electronic, Microelectronic and Nanotechnology, 59652 Villeneuve-d’Ascq, France

**Keywords:** GaN, HEMT, AlGaN back barrier, DIBL, load-pull, PAE, output power density

## Abstract

In this paper, we report on an enhancement of mm-wave power performances with a vertically scaled AlN/GaN heterostructure. An AlGaN back barrier is introduced underneath a non-intentionally doped GaN channel layer, enabling the prevention of punch-through effects and related drain leakage current under a high electric field while using a moderate carbon concentration into the buffer. By carefully tuning the Al concentration into the back barrier layer, the optimized heterostructure offers a unique combination of electron confinement and low trapping effects up to high drain bias for a gate length as short as 100 nm. Consequently, pulsed (CW) Load-Pull measurements at 40 GHz revealed outstanding performances with a record power-added efficiency of 70% (66%) under high output power density at V_DS_ = 20 V. These results demonstrate the interest of this approach for future millimeter-wave applications.

## 1. Introduction

In recent decades, remarkable progress has been achieved with GaN high electron mobility transistors (HEMTs) for use in high frequency power amplification and switching applications. Recent progress has allowed the demonstration of high RF performance up to Ka-band [1,2,3,4,5,6]. However, at a higher frequency, the efficiency and robustness of GaN HEMTs, especially under high drain bias above 15 V, are still limited due to enhanced trapping effects, reduced electron confinement and self-heating when downscaling the device size. Currently, the most matured GaN HEMTs are based on a AlGaN/GaN heterostructure [6,7,8,9,10]. More recently, Al-rich ultrathin sub-10 nm barrier heterostructures have received much attention for millimeter-wave applications [11,12,13,14,15,16,17,18,19,20,21,22]. This is because they can deliver significantly higher 2DEG sheet carrier density compared to AlGaN/GaN HEMTs while offering the possibility to highly scale the epitaxial structure as needed when using short gate lengths [23,24,25]. Therefore, further reducing the gate length to reach a higher frequency of operation requires significant changes of standard epitaxial materials and device design, such as self-aligned gates and an AlGaN back barrier [26,27]. Breakthrough technologies are needed to achieve simultaneously high efficiency under high output power, together with high reliability. A number of new device designs have been developed in this frame, including graded channel HEMTs [28,29] and N-polar HEMT [30,31,32] showing attractive performances in Ka-band and above. In order to further push the performance limits of mm-wave ultrashort GaN transistors, the electron confinement and trapping effects under a high electric field must be optimized. This requires extensive buffer engineering. Recently, we evaluated the impact of various carbon-doping concentrations into the buffer with different undoped GaN channel thickness on the device performance [33]. It was shown that a thin GaN channel, typically below 150 nm, combined with a high carbon concentration into the buffer leads not only to a high electron confinement under high drain bias for 100 nm GaN transistors, but also low leakage current at the expense of trapping effects.

In this work, we insert a thin AlGaN back barrier between a moderately carbon-doped buffer and an undoped GaN channel with the aim of overcoming the trade-off between the electron trapping and the electron confinement, enabling superior bias operation and performances for 100 nm AlN/GaN HEMTs. An extensive Al-content variation has been performed in the back barrier with the aim of optimizing the related polarization and preventing punch-through effects under a high electric field.

## 2. Device Technology

Figure 1a shows a schematic cross-section of HEMT structure grown by metal organic chemical vapor deposition (MOCVD) on 4 in. SiC substrates. A total of four structures have been realized, consisting of an AlN nucleation layer and a 5 × 10^18^ cm^−3^ carbon-doped GaN buffer allowing significantly reduced trapping effects when located away from the channel [33]. This is followed by a 100 nm AlGaN back barrier layer with an Al-content varying from 4% to 25% in order to evaluate the impact on the electron confinement. A 150 nm undoped GaN channel and a 3 nm AlN barrier layer are then used to benefit from both a high polarization and a favorable aspect ratio gate length to gate-to-channel distance in order to mitigate the short channel effects with short gate lengths. Finally, the structures were capped with a 6 nm in situ SiN layer to passivate the surface states and thus avoid DC to RF dispersion.

The source drain ohmic contacts have been formed by a Ti/Al/Ni/Au metal stack annealed at 850 °C directly on top of the AlN barrier by etching the in situ Si_3_N_4_ layer. Ni/Au T-gates with various gate lengths from 100 nm to 250 nm were then defined by e-beam lithography. Figure 1b shows a Transmission Electron Microscopy (TEM) view of the HEMT structure depicting the epitaxial stack as well as the ohmic and Schottky contacts. A 200 nm PECVD Si_3_N_4_ layer was deposited as final passivation. Hall measurements at room temperature show a 2DEG concentration n_s_ ∼ 2 × 10^13^ cm^−2^ with an electron mobility ∼ 950 cm^2^ V^−1^·s^−1^ and a sheet resistance R_sheet_∼ 300 Ω/◻, which is similar for all structures.

The structure variation of the Al-content from 4% to 25% in the AlGaN back barrier is labelled as follows: Al-4%, Al-10%, Al-18% and Al-25%. The energy band diagrams of the structures appear in Figure 1c, showing the increased polarization resulting from a higher Al-content into the back barrier.

## 3. DC and RF Characterization

DC measurements have been performed using a Keysight A2902A static modular and source monitor. Figure 2 shows the output and transfer characteristics of each structure, with a compliance fixed at 150 mA/mm and swept from V_DS_ = 2 V up to 20 V in order to extract the drain bias-induced barrier lowering (DIBL), which is a key parameter for evaluating the 2DEG electron confinement. This required 2 × 50 µm transistors with a gate length of 100 nm and a gate-to-drain distance (L_GD_) of 0.5 µm to be measured on the different structures, Al-4%, Al-10%, Al-18% and Al-25%. The output characteristics show a maximum rain current of approximately 1 A/mm on all structures, which confirms the similarity of the 2DEG density. For the structure Al-4%, we observed a severe degradation of the electron confinement, which is reflected by a large DIBL of 600 mV/V as well as a strong increase of the off-state leakage current. The degradation of the electron confinement is attributed to the low Al-content of 4% in the AlGaN back barrier, for which the back polarization is not high enough to prevent electron injection into the buffer layers under such a high electric field. Therefore, similar structures with a higher Al-content in the back barrier have been tested in order to enhance the electron confinement with short gate lengths. The structure Al-10% indeed shows a drastic improvement of the electron confinement, by a factor of 4, in terms of the DIBL parameter decreasing from 600 mV/V for the structure Al-4% to 130 mV/V for the structure Al-10%. As expected, further increasing the Al-content in the AlGaN back barrier up to 18% and 25% results in a gradual enhancement of the electron confinement, especially the Al-25% structure with a much lower DIBL of less than 30 mV/V for 100 nm gate lengths while maintaining low drain leakage current and high robustness up to V_DS_ = 20 V.

It can be noted that larger gate lengths were measured in the same way. Figure 3a depicts the DIBL as a function of the gate length for each structure. The systematic degradation of the electron confinement is clearly confirmed with short gate lengths (sub-150 nm) when using low Al-content in the AlGaN back barrier (e.g., 4% and 10%). On the other hand, the structures with sufficient Al-content (e.g., >18%) show excellent electron confinement down to 100 nm gate length while maintaining low leakage current and high robustness under a high electric field (V_DS_ > 15 V).

Pulsed I_D_–V_DS_ characteristics at room temperature revealing the current collapse when using various quiescent bias points appear in Figure 3b. The specific pulsed I–V protocol based on I–V characteristics has been settled with the following quiescent bias points: cold point at (V_GQ_ = 0 V, V_DQ_ = 0 V), gate lag at (V_GQ_ = −4 V, V_DQ_ = 0 V) and drain lag at (V_GQ_ = −4 V, V_DQ_ = 10 V, 15 V and 20 V) using a pulsed width of 1 μs and a duty cycle of 1%. This required 2 × 50 µm transistors with L_G_ of 100 nm and L_GD_ of 0.5 µm to be measured, which showed low trapping effects owing to the moderate carbon concentration of 5 × 10^18^ cm^−2^ located 250 nm away from the 2DEG as well as the excellent electron confinement. Indeed, a correlation between an enhanced electron confinement owing to the use of a back barrier and a reduction of electron trapping has been demonstrated [34].

Figure 4 shows S-parameters measured on the structure Al-25% from 250 MHz to 67 GHz using a Rhode and Schwarz ZVA67GHz network analyzer. The current gain extrinsic cut-off frequency (F_T_) slightly decreases with V_DS_ as expected. On the other hand, the maximum oscillation frequency (F_max_) increases as a function of V_DS_, which further confirms the reduced trapping (Figure 4b). F_T_/F_max_ of 60/270 GHz is achieved at V_DS_ = 20 V for a 2 × 50 µm with L_G_ of 100 nm and L_GD_ of 0.5 µm (Figure 4a). The small signal power gain measured at 40 GHz is as high as 17 dB.

## 4. Large Signal Characterization

In order to further validate the benefit of low trapping effects combined with an excellent electron confinement under a high electric field, continuous-wave (CW) and pulsed Load-Pull measurements have been performed on 100 nm transistors from the structure A1-25% at 40 GHz. Details of the power bench used for these measurements can be found in [35]. Figure 5a shows CW PAE and P_OUT_ as a function of the injected power (P_inj_) of a 2 × 50 µm AlN/GaN transistor with L_G_ = 100 nm and L_GD_ = 0.5 µm measured in deep class AB at V_DS_ = 20 V and 25 V. A state-of-the-art PAE above 65% associated with a P_OUT_ of 3.5 W/mm is obtained for an optimum matching PAE. At V_DS_ = 25 V, P_OUT_ increases as expected up to 4.8 W/mm with a peak PAE of 57.5%. It can be noted that the drop of PAE as a function of V_DS_ is mainly due to self-heating.

Figure 5b shows the power performances using the same device in pulsed mode (pulse width of 1 μs and duty cycle of 1%). An outstanding PAE of 70% is measured with a corresponding P_OUT_ of 4.2 W/mm at V_DS_ = 20 V. This heterostructure enables superior drain bias operation up to V_DS_ = 30 V while maintaining high PAE of approximately 60% under a significant P_OUT_ of 7.1 W/mm.

Figure 6a shows a summary of the power performance as a function of drain bias, revealing a rather small gap between CW and pulsed mode. This translates the low trapping effects and enhanced electron confinement under high drain bias at 40 GHz. Figure 6b depicts a benchmark of GaN HEMTs representing the PAE as a function of P_OUT_ from Ka to Q-band. Both CW and pulsed RF power performances appear to be favorably comparable to state-of-the-art GaN HEMTs, reflecting the significant interest in this technology for millimeter-wave applications. For instance, to the best of our knowledge, a PAE > 65% combined with a P_OUT_ > 3 W/mm at 40 GHz in CW is demonstrated for the first time.

## 5. Conclusions

In this work, we evaluated the insertion of a thin AlGaN back barrier with an extensive Al-content variation (from 4% to 25%), with the aim of pushing the limits in terms of drain bias operation of AlN/GaN HEMTs with short gate lengths (down to 100 nm) while maintaining low trapping effects. A higher Al-content in the AlGaN back barrier shows an excellent electron confinement together with low trapping effects despite the significant electric field generated by the short gate length under high drain bias (>20 V). Large signal performances at 40 GHz revealed state-of-the-art power performances for the structure using 25% Al-content in the AlGaN back barrier combined with a moderately carbon-doped GaN buffer. This technology paves the way for highly efficient mm-wave GaN HEMTs delivering superior output power density as needed for next-generation RF power devices.

## Figures and Tables

**Figure 1 micromachines-14-00291-f001:**
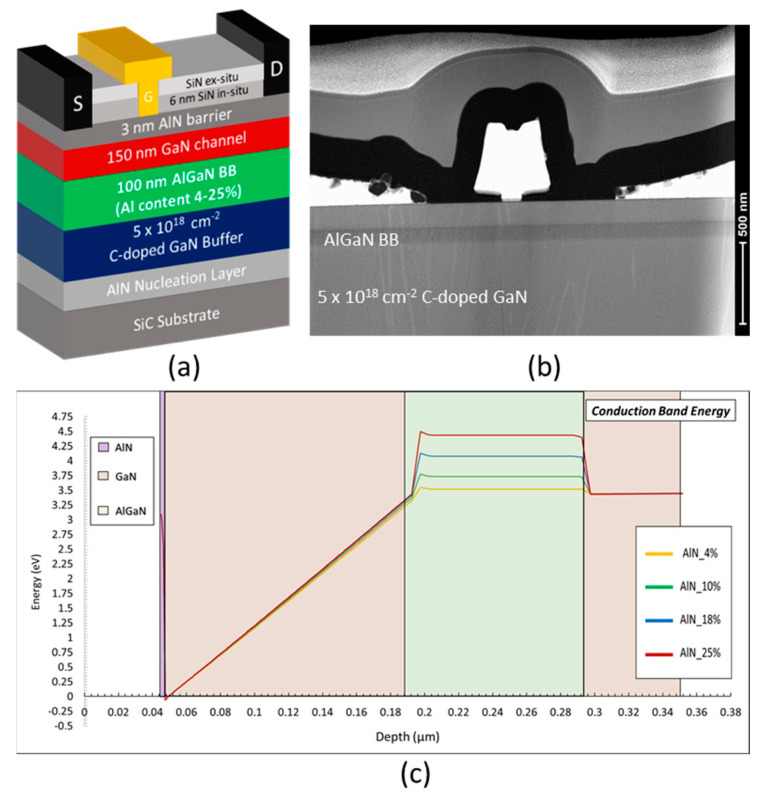
Schematic cross-section of the AlN/GaN HEMT structure based on a thin AlGaN back barrier with various Al-content (4% to 25%) (**a**), TEM view of the device showing the epitaxial stack as well as the ohmic and Schottky contacts (**b**) and the energy band diagram of the structures with different Al-content (**c**).

**Figure 2 micromachines-14-00291-f002:**
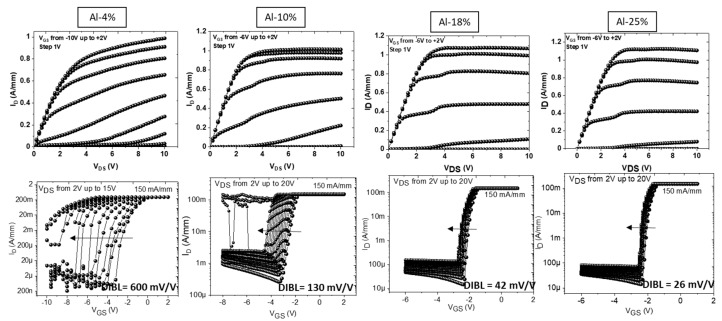
Output and transfer characteristics from V_DS_ = 2 V up to 20 V of 2 × 50 µm AlN/GaN HEMTs for L_GD_ = 0.5 µm and L_G_ = 100 nm with various Al-content in the AlGaN back barrier.

**Figure 3 micromachines-14-00291-f003:**
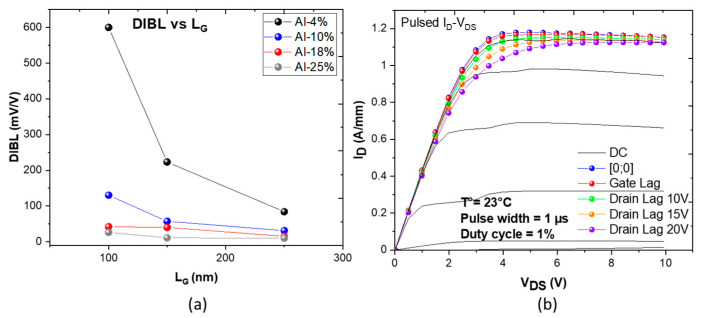
DIBL as a function of the gate length for AlN/GaN HEMTs with various Al-content in the AlGaN back barrier (**a**) and open channel pulsed I_D_–V_DS_ output characteristics of 2 × 50 µm AlN/GaN HEMT (Al-25%) with L_G_ = 100 nm and L_GD_ = 0.5 µm (**b**).

**Figure 4 micromachines-14-00291-f004:**
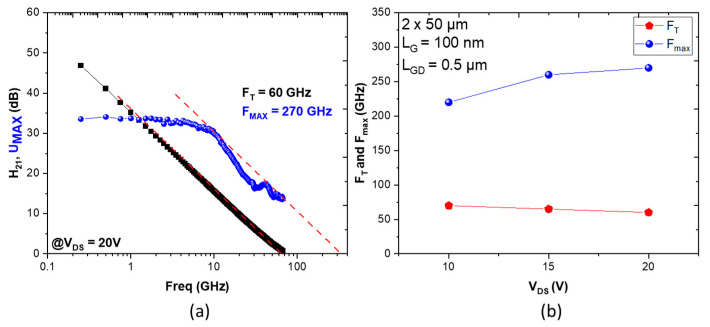
(**a**) F_T_/F_max_ at V_DS_ = 20 V and (**b**) F_T_/F_max_ as a function of V_DS_ for a 2 × 50 µm AlN/GaN HEMT (Al-25%) with L_G_ = 100 nm and L_GD_ = 0.5 µm.

**Figure 5 micromachines-14-00291-f005:**
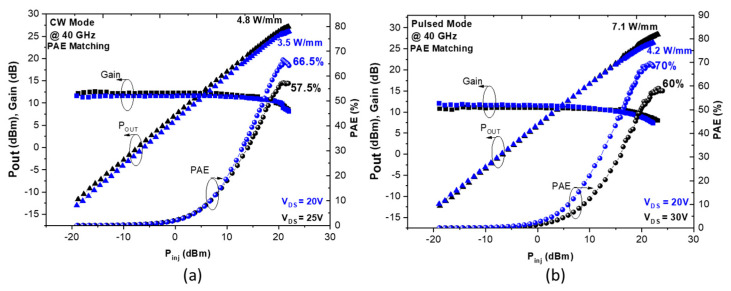
Typical large signal performances at 40 GHz for a 2 × 50 µm AlN/GaN HEMT (Al-25%) with L_G_ = 100 nm and L_GD_ = 0.5 µm in CW mode up to 25 V (**a**) and pulsed mode up to 30 V (**b**).

**Figure 6 micromachines-14-00291-f006:**
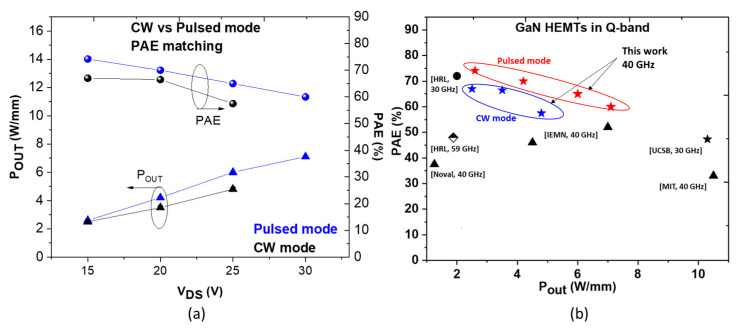
CW and pulsed P_OUT_ and PAE as a function of V_DS_ at 40 GHz for AlN/GaN HEMT (Al-25% structure) (**a**) and PAE versus P_OUT_ benchmark of GaN HEMTs in Ka and Q-band (**b**).

## Data Availability

Not applicable.

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
