# Peer review of "Low Trapping Effects and High Electron Confinement in Short AlN/GaN-On-SiC HEMTs by Means of a Thin AlGaN Back Barrier"

_micromachines, 2023, doi:10.3390/mi14020291_

Round 1

Reviewer 1 Report

The main contribution of this work is insert a thin AlGaN back barrier between a moderately carbon doped buffer and an undoped GaN channel. And their achievements are demonstrated by the measures shared in this work.

Figure 5(b) does not sufficiently highlight the work situation. It would be interesting to add a comparison with other works where the improvements made in your work can be appreciated more clearly.

Have no self-heating effects of the device been observed?

Reviewer 3 Report

This is a well-written article on a timely subject, in which the authors study the compromise between electron confinement and trapping degradation in GaN HEMTs for millimeter-wave applications. Experimental data are provided, which makes it a valuable contribution. However, there are a few suggestions for improvement:

- In the introduction/abstract, it would be helpful to more clearly state the focus/novelty of this work. AlGaN back-barrier and carbon-doping are known for this technology, so it would be useful to provide a comparison table with the actual numbers from other technological variants reported in the literature. Is the focus of this work the analysis of different Al-content in the back barrier?

- It would be interesting to see more studies at different Al concentrations, for example the dependency of gate/drain-lag effects and large-signal performance. Only DIBL is studied in this sense, whereas all the other characterization data are for the Al-25%. Do the authors have data or comments to add?

- Trapping effects are often studied by Deep-Level Transient Spectroscopy. The reference A. M. Angelotti, et al., "Experimental Characterization of Charge Trapping Dynamics in 100-nm AlN/GaN/AlGaN-on-Si HEMTs by Wideband Transient Measurements," in IEEE T-ED 2020 should be added, as the same technology is characterized and evaluated in terms of trapping. Do the authors have similar characterization to show in this case?

- Self-heating is mentioned in the RF characterization. Could the authors provide data on the thermal resistance and have an idea of the thermal constants involved? It would also be helpful to discuss the mutual interaction with trapping.

- It would be useful to report the output characteristics (Id vs Vds) for the data in Fig. 2.

- In the caption of Fig. 3, the explicit reference to (b) is missing.

- While S-parameters are mentioned, they are not reported. It would be useful to have the plots in this paper.

- Overall, these suggestions would make the article more complete and add to its value.

Round 2

Reviewer 3 Report

Thanks for replying to the comments and for modifying the manuscript accordingly.

Author Response

Thank you for your time and valuable comments.